# Caloric Restriction Mimetic 2-Deoxyglucose Reduces Inflammatory Signaling in Human Astrocytes: Implications for Therapeutic Strategies Targeting Neurodegenerative Diseases

**DOI:** 10.3390/brainsci12030308

**Published:** 2022-02-24

**Authors:** Kaylie-Anna Juliette Vallee, Jerel Adam Fields

**Affiliations:** Department of Psychiatry, University of California, San Diego, CA 92093, USA; kvallee@health.ucsd.edu

**Keywords:** neurodegenerative disease, caloric restriction, glycolysis, astrogliosis, inflammation, immunometabolism, 2-deoxyglucose

## Abstract

Therapeutic interventions are greatly needed for age-related neurodegenerative diseases. Astrocytes regulate many aspects of neuronal function including bioenergetics and synaptic transmission. Reactive astrocytes are implicated in neurodegenerative diseases due to their pro-inflammatory phenotype close association with damaged neurons. Thus, strategies to reduce astrocyte reactivity may support brain health. Caloric restriction and a ketogenic diet limit energy production via glycolysis and promote oxidative phosphorylation, which has gained traction as a strategy to improve brain health. However, it is unknown how caloric restriction affects astrocyte reactivity in the context of neuroinflammation. We investigated how a caloric restriction mimetic and glycolysis inhibitor, 2-deoxyglucose (2-DG), affects interleukin 1β-induced inflammatory gene expression in human astrocytes. Human astrocyte cultures were exposed to 2-DG or vehicle for 24 h and then to recombinant IL-1β for 6 or 24 h to analyze mRNA and protein expression, respectively. Gene expression levels of proinflammatory genes (complement component 3, IL-1β, IL6, and TNFα) were analyzed by real-time PCR, immunoblot, and immunohistochemistry. As expected, IL-1β induced elevated levels of proinflammatory genes. 2-DG reversed this effect at the mRNA and protein levels without inducing cytotoxicity. Collectively, these data suggest that inhibiting glycolysis in human astrocytes reduces IL-1β-induced reactivity. This finding may lead to novel therapeutic strategies to limit inflammation and enhance bioenergetics toward the goal of preventing and treating neurodegenerative diseases.

## 1. Introduction

The world’s aging population is susceptible to age-related neurodegenerative diseases, for many of which no effective preventive therapies or treatments are available. Neuroinflammation is a hallmark of many neurodegenerative diseases and represents a potential therapeutic target to improve the health and quality of life of aging populations [1,2]. Microglia and astrocytes produce and perpetuate neuroinflammation in the brain and are believed to contribute to neurodegeneration when in a chronically reactive state [3,4,5,6,7]. Growing evidence suggests that modulating cellular metabolism may represent a way to target inflammatory signaling [8,9,10,11,12,13]. However, few studies have examined whether altering metabolic signaling in astrocytes affects inflammatory gene expression.

Developing a therapeutic strategy that disrupts inflammatory signaling and restores bioenergetic homeostasis in the brain would be highly beneficial to people suffering from disorders of the central nervous system. Recent findings by our group and others suggest that proinflammatory stimuli cause mitochondrial dysfunction in neurons [8,9,10,11,12,13]. We have discovered evidence that reactive astrocytes cause a reduction in the levels of transcription factor at mitochondria (TFAM) in neurons that results in deficient mitochondrial biogenesis [8,9]. Blocking metabolic activity in reactive astrocytes with an anti-inflammatory cannabinoid receptor agonist (WIN55,212-2 [WIN]) proved neuroprotective in an in vitro model for neurons [8]. However, WIN blocked both glycolytic and oxidative metabolism. Thus, it is unknown if inhibiting glycolysis alters the inflammatory and metabolic phenotype of astrocytes.

Astrocytes play a multifactorial role in maintaining brain homeostasis, including but not limited to energy substrate procurement from the blood, regulation of neurotransmitter concentrations, blood–brain barrier integrity, and responding to pathogenic stimuli [6,7,14]. Emerging data indicate a complex coupling of metabolic changes to these processes in astrocytes. Under homeostatic conditions, astrocytes are primarily glycolytic, secreting lactate as a byproduct to the extracellular space [15,16]. Neurons import this lactate from the extracellular space and then metabolize it in mitochondria via oxidative phosphorylation to generate ATP [15,16]. Seminal studies have shown that the astrocyte-to-neuron lactate shuttle may be important for memory formation and synaptic plasticity, processes that are implicated to be disrupted in HAND [15,17]. Reactive astrocytes transiently utilize lactate in OXPHOS to fuel responses to infection and injury [4,5,6,7,8,9,10]. Acutely, this metabolic switch in astrocytes serves to restore homeostasis, but in chronic conditions of neuroinflammation, this increase in astrocyte mitochondrial activity may deprive neurons of lactate, resulting in an energy deficit [10,11,12]. However, recent studies suggest that the astrocyte-to-neuron lactate shuttle is not as crucial to brain function as once thought [18,19]. There is contrary evidence that neurons are primarily fueled by glucose that is taken up from the extracellular space and metabolized sequentially via glycolysis and oxidative phosphorylation [18,19]. Alternatively, there is evidence that neurons can maintain energy levels and neuronal function in low-glucose environments, such as caloric restriction or ketogenic dieting, via fatty acid oxidation [20,21,22,23]. However, the metabolism of astrocytes in low-glucose conditions is not well studied. Despite these findings, which metabolic substrates and downstream processes are most important for fueling astrocyte reactivity is not known.

Chronically reactive astrocytes secrete inflammatory cytokines and may alter blood–brain barrier permeability as well as the uptake of neurotransmitters from extracellular space in a way that contributes to the neurodegenerative process [3,4,5,6,7,14,24,25]. Upon immune stimulation, astrocytes produce inflammatory cytokines and other inflammatory genes including the complement component 3 (C3) [5,26,27]. Transiently, these changes in gene expression serve to restore homeostasis, but as with concomitant metabolic changes, chronic increases in the expression of inflammatory genes may be detrimental to brain health. Experimental evidence has shown that upregulated levels of C3 can promote the neuroinflammatory phenotype of reactive astrocytes and has a role in the pathogenesis of various neurodegenerative diseases, including AD. Astrocytes also regulate synaptic transmission by taking glutamate up from the extracellular space surrounding synapses, metabolizing the glutamate to glutamine, and returning the glutamine to the extracellular space for uptake by neurons [28]. Reactive astrocytes have reduced capacity to take up glutamate, and glutamate excitotoxicity may contribute to neurodegeneration [29,30,31]. However, it is unknown how astrocyte metabolism may affect inflammatory gene expression and glutamate regulation. 

Caloric restriction and ketogenic diets have been shown to improve brain function. In rodent models, caloric restriction was found to extend the life span and reduce the occurrence of age-related diseases [32]. Ketogenic diets have long been used to treat pediatric epilepsy [20,23]. β-hydroxybutyrate, a by-product of ketosis, has been shown to improve neuronal function [33]. Studies have shown that β-hydroxybutyrate may be neuroprotective by mimicking caloric restriction and reducing inflammation [34]. Despite these findings, little is known about how astrocytes respond to caloric restriction or a ketogenic diet, both of which limit reliance on glycolysis for ATP. 

In this study, we aimed to determine if inhibiting glycolysis in astrocytes alters their response to inflammatory cytokines. We found that the caloric restriction mimetic and glycolytic inhibitor 2-deoxyglucose inhibits IL-1β-induced inflammatory gene expression in a dose-dependent manner. These data suggest that reducing the reactivity of astrocytes may be a mechanism through which caloric restriction and ketogenic diets are beneficial to the brain.

## 2. Materials and Methods

### 2.1. Generation of Human Astrocytes 

This study was approved by the University of California San Diego Human Research Protections Program and deemed IRB exempt (Federalwide Assurance #00000021 and Institutional Review Board #IORG0000210 [7 March 2019]). Astrocytes used in this study were from a differentiated cell line originally generated prior to 5 June 2019 (as per NIH NOT-OT-19-128) from fetal human brain tissue from terminated pregnancy between 12 and 16 weeks of gestation, as previously described [35]. Donors gave written informed consent for research use of the cells and tissue. Tissue was fragmented and mechanically dissociated using a scalpel and washed 3 times with a HBSS holding medium (Gibco™, Waltham, MA, USA, cat. no.14175-095) with 1 mM Glutamax (Gibco, cat. no. 35050-061), 20 μg/mL Gentamicin (Gibco, cat. no. 15710-064) and 5 mM HEPES (Gibco, cat. no. 15630-080). Tissue was homogenized with the addition of 15 mL of 0.25% trypsin EDTA (Gibco, cat. no. 25200-056) for 5 min in a 37 °C incubator. After 5 min, 1 mL of a trypsin inhibitor (Roche Diagnostics, Indianapolis, IN, USA, cat. no. 10109) and 24 mL of DMEM (Gibco, cat. no. 11960-044) with human serum (Corning™, Corning, NY, USA, cat. no. 35-060-cl) were added. The mixture was then centrifuged for 5 min at 4 °C to pellet the cells. Supernatant was removed and discarded, and the cells were resuspended in 5 mL of DMEM and strained with a 70 μM strainer (Corning, Falcon^®^, Durham, NC, USA, cat. no. 352350). The cell suspension was underlaid with 7 mL of a solution of filtered 8% BSA in PBS and cells were centrifuged at 1 × 10^4^ rpm at 4 °C for 10 min. The supernatant was removed, and the cells were resuspended in DMEM with human serum for the astrocyte medium (Gibco, cat. no. 21103-049)), 1 mM GlutaMAX, and 20 μg/mL Gentamicin. Astrocytes were plated at a density of 1 × 10^7^/T75 flask and cultured as adherent monolayers. After 1 week, the astrocyte medium with human serum was replaced with DMEM with 10% fetal bovine serum (FBS) (Gibco, cat. no. 16000044), 1% penicillin/ streptomycin (P/S) (Corning, cat. no. 30-001-CI-1), and 1% L-glutamine (Gibco, cat. no. 25030-081). Every 3 days, a half medium exchange was performed. The same donor line was used for all experiments with different passages being used throughout this study.

### 2.2. Treatment of Astrocytes 

Astrocytes were cultured in 12-well plates at 500,000 cells/well on the day prior to treatment. To mimic caloric restriction and inhibit glycolysis, astrocytes were treated with increasing doses of 2-DG (1, 5, 10, and 20 mM [50 mM concentration was included in the cytotoxicity assay]) for 24 h. To determine if presence of pyruvate, necessary for oxidative phosphorylation in the absence of glycolysis, affects 2-DG inhibition of IL-1β-induced gene expression in astrocytes, parallel experiments were conducted using the medium without pyruvate. To model an inflammatory environment, astrocytes were treated with IL-1β for 6 h at a concentration of 10 ng/mL, a concentration consistent with the relevant literature [36]. To assess the effects of 2-DG on expression levels of multiple IL-1β-induced inflammatory genes, astrocytes were pretreated with 2-DG at 20 mM or vehicle for 24 h and then treated with IL-1β or vehicle for 6 h prior to RNA isolation for analysis by RT-qPCR. 

### 2.3. RNA Isolation and TaqMan^®^ Human Inflammation Array and Real-Time Reverse Transcription Polymerase Chain Reaction (RT^2^PCR) 

Following the 6 h treatment with IL-1β (10 ng/mL) on astrocytes cultured in 12-well plates, the medium was removed from the wells and cells were washed once with PBS. RNA was extracted using the RNeasy plus mini kit (Qiagen, Germantown, MD, USA, cat. no. 74136) according to the manufacturer’s instructions. A spectrophotometer was used to analyze the purity and concentration of RNA samples. For the RT^2^PCR, RNA was reverse transcribed into cDNA with a high-capacity cDNA Reverse Transcription Kit (Life technologies™, Waltham, MA, USA, cat. no. 4358813) per the manufacturer’s instructions. Gene expression was determined using RT^2^PCR TaqMan gene expression assays with the QuantStudio 3 sequence-detections system (Life Technologies^TM^ ) using Taqman primers specific to IL-1β (cat. no. Hs01555410_m1), IL6 (cat. no. Hs00174131_m1), TNFα (cat. no. Hs00174128_m1), C3 (cat. no. Hs00163811_m1), and LCN2 (cat. no. Hs01008571_m1). An ActB (Applied Biosystems™, Waltham, MA, USA, cat. no. 1612290) primer was used as a normalization control. A master mix was created using 2 × fast advanced master mix (Thermo Fisher Scientific, Waltham, MA, USA, cat. no. 4444557), 20 × primers, and water. Each reaction well of a microamp fast optical plate (Applied Biosystems, cat. no. 4346907) received 8.5 µL of the master mix and 1.5 µL cDNA. The reactions were carried out at 48 °C for 30 min, 95 °C for 10 min, followed by 40 cycles of 95 °C for 15 s and 60 °C for 1 min. Each sample was analyzed in duplicate and their C_T_ values were collected, exported to an Excel file, and used to calculate fold changes using the comparative C_T_ method [37].

### 2.4. Immunoblot 

Cells were plated on 12-well plates at 300,000 or 500,000 cells per well to isolate RNA and protein, respectively, and treated for 24 h with vehicle, IL-1β 10 ng/mL, 2-DG 20 mM, or IL-1β 10 ng/mL and 2-DG 20 mM. To determine a dose-response to 2-DG, cells either received treatment for vehicle, IL-1β 10 ng/mL, increasing doses of 2-DG (1, 5, 10, 20, or 50 mM), or IL-1β and increasing doses of 2-DG (1, 5, 10, 20, or 50 mM). Cells were treated with vehicle or 2-DG 24 h prior to treatment with IL-1β or vehicle. On the day following treatment, solution was removed from the cells, and they were washed with sterile PBS. PBS was removed, and cells were lysed using a solution of 0.1% Triton-X in PBS with the addition of protease inhibitors. Cells were then centrifuged at 2000 rpm for 5 min to obtain the whole protein lysate. After the protein concentration was determined using bicinchoninic acid assay (Thermo Fisher Scientific, cat. no. 23225), the samples were denatured in lamellae sample buffer (Bio-Rad, Hercules, CA, USA, cat. no. 1610747). Whole-lysate samples were loaded (10 μg total protein/lane) on 4–15% Criterion TGX stain-free gels (Bio-Rad, cat. no. 5678085), electrophoresed in tris/glycine/SDS running buffer (Bio-Rad, cat. no. 161-0772), and transferred onto an LF PVDF membrane with Bio-Rad transfer stacks and transfer buffer (Bio-Rad, cat. no 1704275) using the Bio-Rad Trans Blot Turbo transfer system. After the transfer, total protein was imaged using a Bio-Rad ChemiDoc imager under the stain-free blot setting for normalization purposes. The membranes were then blocked in 1% casein in tris-buffered saline (TBS) (Bio-Rad, cat. no. 1610782) for 1 h. Membranes were incubated overnight at 4 °C with primary antibodies, C3 1:500 (Santa Cruz Biotechnology, Inc, Dallas, TX, USA, cat. no. sc-20137) diluted in blocking buffer. All blots were then washed in PBS-T, and incubated with species-specific IgG conjugated to HRP (American Qualex, San Clemente, CA, USA, cat. no. A102P5) diluted 1:5000 in PBS-T and visualized with SuperSignal West Femto Maximum Sensitivity Substrate (Thermo Fisher Scientific, cat. no. 34096). Images were obtained, and semi-quantitative analysis was performed with the ChemiDoc gel imaging system and Quantity One software (Bio-Rad). All experiments were performed in biological replicates of three and repeated in three independent experiments.

### 2.5. Quantification of Astrocyte Nuclei

To determine cell numbers after exposure for 24 h to 2-DG, astrocytes were cultured in 96-well plates and treated as described above (Section 2.2). After treatment with 2-DG, astrocytes were fixed using 4% paraformaldehyde for 20 min, washed three times with phosphate-buffered saline (PBS) and then exposed to 2-(4-crbmimidolphenyl)-1H-indole-6-carmoximidamide, dihydrochloride (DAPI) for 10 min to stain nuclei. Astrocytes were washed three times with PBS and then visualized using a fluorescent microscope. Five random images were acquired in each of three wells and DAPI-positive nuclei were counted in each field of view and quantities compared between treatment groups. Experiments were performed in triplicate and repeated three times.

### 2.6. Cytotoxicity Assay

To determine the cytotoxicity of 2-DG on human astrocytes, cells were cultured in 96-well white tissue culture plates and treated as described above (Section 2.2) and analyzed for cytotoxicity using the CytoTox-Glo Cytotoxicity Assay (Promega, Madison, WI, USA, cat. no. G9290) as per the manufacturer’s instructions. Luminescence was measured using a BioTek multiplate reader. A positive control (lysis buffer) for cell death was used to calculate the percent cell death in each well and quantities compared between treatment groups. Experiments were performed in triplicate and repeated three times.

## 3. Results

### 3.1. 2-Deoxyglucose Blocks IL-1β-Induced Inflammatory mRNA, IL6, in a Dose-Dependent Manner 

To investigate how reactive astrocytes respond when glycolysis is inhibited, we used IL-1β to stimulate astrogliosis and the glycolytic inhibitor 2-deoxyglucose (2-DG) to experimentally inhibit glycolysis. Primary human astrocyte cultures were pretreated with increasing concentrations of 2-DG followed by treatment with IL-1β (10 ng/mL). RNA was extracted and analyzed by RT^2^ PCR to determine the expression of the inflammatory gene IL6. As expected, IL-1β caused a significant increase in IL6 mRNA levels (*** *p* < 0.001) compared to vehicle (Figure 1). Astrocytes pretreated with 2-DG showed reduced IL-1β-induced IL6 mRNA levels in a dose-dependent manner (Figure 1A,B). The 20 mM 2-DG dose was most effective at reducing the IL-1β-induced IL6 mRNA levels by 62% (Figure 1A) and 65% (Figure 1B) (^^^ *p* < 0.001) in astrocytes in media with or without pyruvate. 

### 3.2. 2-Deoxyglucose Blocks IL-1β-Induced Inflammatory mRNA in Human Astrocyte

To understand how 2-DG affects the expression of IL-1β-induced genes, astrocytes were pretreated with 2-DG (20mM) followed by treatment with vehicle or IL-1β (10 ng/mL). The expression of various inflammatory genes, including IL-1β, TNFα, C3, and LCN2, was analyzed by RT^2^PCR. Consistent with previous findings, treatment with IL-1β alone caused a significant increase in the expression of all mRNA transcripts (Figure 2A–D). Pretreatment of 2-DG (20 mM) followed by stimulation with IL-1β (10 ng/mL) reduced the effect of IL-1β-induced mRNA by 64% for IL-1β (Figure 2A), 53% for TNFα (Figure 2B), 66% for C3 (Figure 2C), and 81% for LCN2 (Figure 2D). Collectively, these data suggest that 2-DG blocks IL-1β-induced inflammatory mRNA production and the degree of this effect is likely dependent on the targeted gene. 

### 3.3. IL-1β-Induced C3 Protein Production Hindered by 2-DG in Human Astrocytes

To further investigate the effects of 2-DG on activated C3, astrocytes were pretreated with vehicle or 2-DG (20 mM) followed by treatment with vehicle or IL-1β (10 ng/mL). Isolated protein lysates were prepared for immunoblot to validate the observed effects of 2-DG on IL-1β-induced gene expression at the protein level. As expected, IL-1β-treated cells showed significant increases in band intensity compared to vehicle, with verification by densitometry analysis showing a 5.3-fold increase in C3 levels (Figure 3A,B). Astrocytes treated with 2-DG alone or 2-DG + IL-1β showed lower C3 band signals compared to those treated with IL-1β alone (Figure 1A), with densitometry analysis showing a 92.4% and 51.5% decrease, respectively (Figure 1B; ^^^ *p* < 0.001, ^^ *p* < 0.01). 

### 3.4. 2-DG Is Not Cytotoxic to Human Astrocytes

To determine if the 2-DG-induced reduction in the expression of inflammatory genes by astrocytes is related to cell viability, astrocytes were exposed to increasing doses of 2-DG (1, 5, 10, and 20 mM) for 24 h and then to IL-1β for 6 h before determining cytotoxicity by quantifying nuclei and using the CytoTox Glo assay. Quantification of nuclei using DAPI staining and fluorescent microscopy showed no significant difference in nuclei numbers between vehicle-treated cultures and cultures treated with increasing doses of 2-DG (Figure 4A). Interestingly, compared to vehicle-treated astrocytes, significantly increased luminescence was detected in 2-DG-treated astrocytes in a dose-dependent manner (Figure 4B). 

### 3.5. 2-DG Blocks IL-1β-Induced Gene Expression in Human Astrocytes

Previous studies have shown that IL-1β induces astrocyte inflammatory gene expression to levels that match neuroinflammation seen in various neurodegenerative diseases [8]. 2-deoxyglucose, a caloric restriction mimetic and glycolytic analog, has been found to inhibit glycolysis in other cell models [38]. The accumulated evidence of this study shows that 2-deoxyglucose blocks the IL-1β-induced expression of inflammatory molecules in human astrocytes (Figure 5). 

## 4. Discussion

This study is the first to show that a caloric restriction mimetic, 2-DG, has inhibitory effects on inflammatory signaling and metabolism in human astrocytes. In a dose-dependent manner, 2-DG diminished IL-1β-induced inflammatory gene expression in human astrocytes. Specifically, 2-DG reduced the expression of the inflammatory genes C3, TNFα, LCN2, and IL6, all of which are implicated in various neurodegenerative and neurological diseases. These findings suggest that inflammatory signaling in the brain may be amenable to manipulation via modifying the availability of metabolic substrates. Furthermore, these findings may offer a novel paradigm for therapeutic targeting of neurodegenerative diseases. 

Many neurodegenerative and neurological disorders are associated with metabolic abnormalities [39]. Alzheimer’s disease, Parkinson’s disease, and HIV-associated neurocognitive disorders all feature altered metabolic profiles and mitochondrial abnormalities in the brain compared to age-matched controls [39,40,41,42,43]. Patients with diabetes are at higher risk for developing inflammatory and neurodegenerative-related diseases, suggesting that altered glucose metabolism may predispose to such ailments [44,45,46]. The findings presented here are consistent with studies showing that caloric restriction and ketogenic diets may be anti-inflammatory and neuroprotective in models for neurodegenerative disease. 

Reactive astrocytes, inflammatory cytokines, complement component proteins, and other inflammatory genes are all involved in development and progression of neurodegenerative diseases. Liddelow et al. (2017) found that the complement protein C3 is a marker of reactive astrocytes [4,5]. Moreover, C3 expression is associated with metabolic abnormalities in disease [47]. Our recent results show that IL-1β induces inflammatory gene expression and metabolic changes in astrocytes in ways that are consistent with neurodegenerative diseases [8]. Due to the central role of astrocytes in regulating brain homeostasis by procuring energy substrate from the peripheral blood supply, regulating BBB integrity, and modulating synaptic function, prolonged astrocyte reactivity may compromise brain function. Our recent studies show that in vitro reactive astrocytes are neurotoxic and reducing astrocyte metabolism may be neuroprotective in this context [8]. Thus, reducing the activation of astrocytes may be neuroprotective via restoring proper functioning of astrocyte processes. The findings here showing that 2-DG inhibits IL-1β-induced gene expression in the absence or presence of pyruvate suggests that inhibiting glycolysis is sufficient to reduce inflammatory signaling in astrocytes. Future studies are needed to determine if energy substrates that promote oxidative phosphorylation over glycolysis alter the number and characteristics of reactive astrocytes in animal models specific for neurodegenerative diseases. 

Oxidative stress, altered autophagy, and dysfunctional mitochondrial function are all associated with neurodegenerative diseases [48,49,50,51,52,53,54,55,56,57]. Accumulation of oxidized lipids and proteins is associated with multiple neurodegenerative diseases [58,59,60]. The accumulation of oxidative damage may stem from a combination of mitochondrial dysfunction leading to over production of reactive oxygen species coupled with reduced autophagy, resulting in reduced turnover of damaged lipids, proteins, and organelles. It has been postulated that caloric restriction reduces the amount of glucose available as a substrate and induces ketogenesis, which increases turnover of damaged macromolecules [61,62]. Our data suggest that in addition to these changes, caloric restriction may inhibit inflammation, which is consistent with the neuroprotective effects of caloric restriction observed in models for neurodegenerative diseases.

2-DG has been shown in multiple studies to be toxic to tumor cells and for this reason has been investigated as a therapy against cancer [63]. However, these studies show that 2-DG is not toxic to human astrocyte cultures. This difference in effects on tumor cells and human astrocytes may be due to tumor cell dependence on glycolysis whereas astrocytes are able to switch from glycolysis to oxidative phosphorylation. This mechanism needs to be further investigated in futures studies. 

Despite the exciting findings presented here, this study has several limitations and further investigation is needed. These studies do not determine if 2-DG prevents the neurotoxicity of reactive astrocytes that has been demonstrated in experiments by our group and others. Further, this study was limited in the investigation of a single donor line of human astrocytes and did not evaluate the effects 2-DG has on the relationship between astrocytes and neurons. This study also does not determine in a comprehensive way the gene expression changes that are induced by 2-DG, nor does it sufficiently evaluate these effects at the protein level. As previously mentioned, more studies are needed to determine the effects of caloric restriction and ketogenic diets on astrocytes and neuroprotection in vivo. This study does not measure astrocytic metabolic function, ATP, or lactate levels produced by astrocytes exposed to IL-1β and 2-DG, important molecules to better understand effects on astrocytes. Future studies are being designed to fully understand the potential of 2-DG, caloric restriction, and ketogenic diets to modulate astrocyte function and provide neuroprotection for a wide range of neurodegenerative diseases. 

## 5. Conclusions

These findings lay the groundwork to investigate how energy substrate utilization may alter astrocyte biology, glutamate signaling, neuroinflammation, blood–brain barrier integrity, neurodegeneration, and overall brain health. Considering the widespread occurrence of metabolic disorders and diseases such as diabetes, which are associated with neurological dysfunction, modulating metabolism in a neuroprotective manner may represent a much-needed strategy to improve brain health. Future studies are needed to further determine metabolic strategies to prevent brain dysfunction in individuals and populations. 

## Figures and Tables

**Figure 1 brainsci-12-00308-f001:**
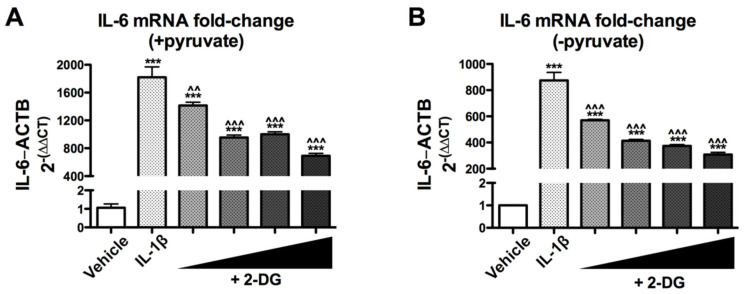
2-DG blocks IL-1β-induced inflammatory gene expression in human astrocytes in a dose-dependent manner. Human astrocytes were pretreated with increasing concentrations of 2-DG (1, 5, 10, and 20 mM) followed by treatment with IL-1β 6 h prior to RNA isolation and gene expression analyses by RT^2^PCR. (**A**) Fold change of IL6 mRNA levels normalized to ACTB mRNA levels in the medium containing pyruvate. (**B**) Fold change of IL6 mRNA levels normalized to ACTB mRNA levels in the medium without pyruvate. One-way ANOVA was performed to the determine effect of treatment on IL6 cultured in media with pyruvate (**A**) [F (9, 10) = 83.52, *p* < 0.0001] and without pyruvate (**B**) [F (9, 10) = 116.1, *p* < 0.0001]. A *post hoc* Tukey’s test was conducted with corrected *p*-values shown (*** *p* < 0.001 vs. vehicle; ^^ *p* < 0.01; ^^^ *p* < 0.001; vs. IL-1β-treated cells). ANOVA, analysis of variance.

**Figure 2 brainsci-12-00308-f002:**
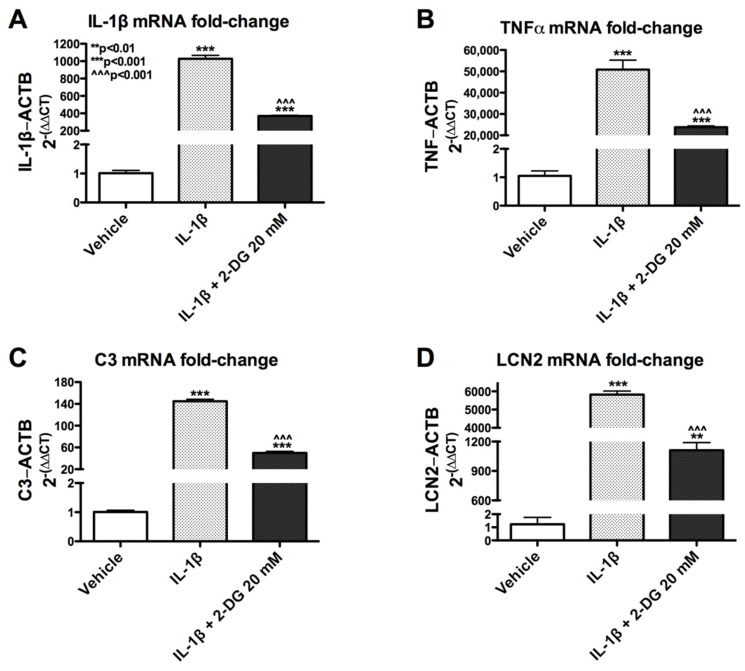
2-DG blocks the expression of multiple IL-1β-induced inflammatory genes in human astrocytes. Fold change of IL-1β (**A**), TNFα (**B**), C3 (**C**), or LCN2 (**D**) mRNA transcript levels normalized to ACTB mRNA levels in total RNA isolated from human astrocytes. One-way ANOVA was performed to the determine effect of treatment on IL-1β [F (9, 10) = 505.5, *p* < 0.0001], TNFα [F (9, 10) = 96.72, *p* < 0.0001], C3 [F (9, 10) = 796.1, *p* < 0.0001], and LCN2 [F (9, 10) = 516.8, *p* < 0.0001]. A *post hoc* Tukey’s test was conducted with corrected *p*-values shown (** *p* < 0.01; *** *p* < 0.001 vs. vehicle; ^^^ *p* < 0.001 vs. IL-1β-treated cells).

**Figure 3 brainsci-12-00308-f003:**
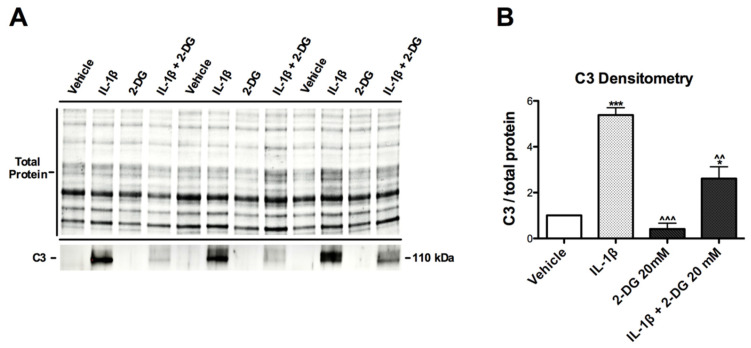
2-DG blocks IL-1β-induced C3 protein in human astrocytes. (**A**) Immunoblot of human astrocyte treated with IL-1β or IL-1β + 2-DG 20 mM with antibody specific for C3, normalized to total protein. (**B**) Quantification of C3 band intensity by IL-1β vs. IL-1β + 2-DG 20mM. Statistical significance was determined by an unpaired *t*-test. * *p* < 0.05, *** *p* < 0.001 vs. vehicle; ^^ *p* < 0.01, ^^^ *p* < 0.001 vs. IL-1β.

**Figure 4 brainsci-12-00308-f004:**
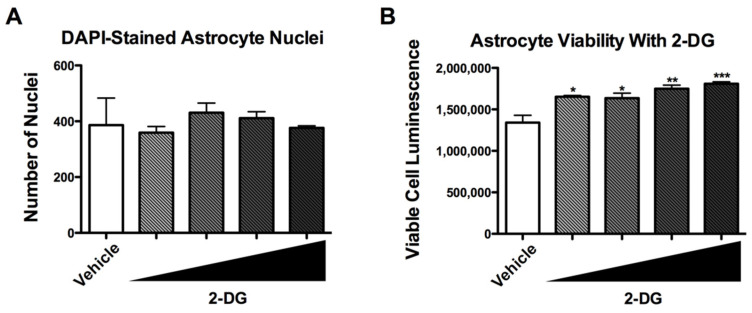
2-DG does not contribute to cell death in astrocytes. (**A**) Fluorescence imaging and quantification of nuclei present in astrocyte cultures treated with increasing concentrations of 2-DG (1, 5, 10, and 20 mM). (**B**) Viable cell luminescence values of astrocytes treated with increasing doses of 2-DG (1, 5, 10, and 20 mM). One-way ANOVA was performed to the determine effect of 2-DG treatment on astrocyte viability [F (9, 10) = 11.72, *p* < 0.001]. A *post hoc* Tukey’s test was conducted with corrected *p*-values shown (* *p* < 0.05; ** *p* < 0.01; *** *p* < 0.001 vs. vehicle).

**Figure 5 brainsci-12-00308-f005:**
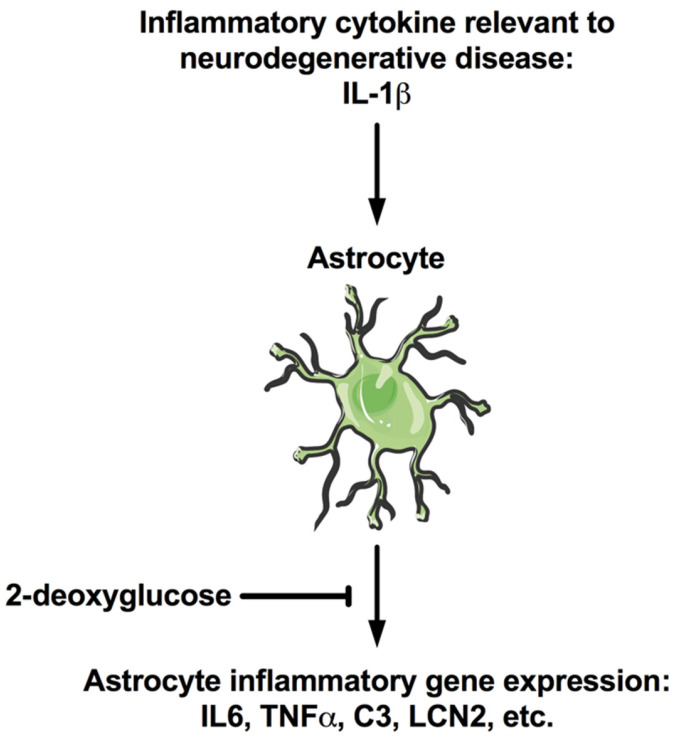
2-DG blocks IL-1β-induced inflammatory gene expression in human astrocytes. IL-1β is a prototypic inflammatory cytokine that is relevant to many neurodegenerative diseases. IL-1β was used here to model neuroinflammation while 2-DG was used to mimic caloric restriction, a phenomenon relevant to fasting and exercise, both of which may have beneficial effects on the brain. Our findings indicate that 2-DG reduces the IL-1β-induced expression of inflammatory genes in astrocytes including, but not limited to, IL6, TNFα, C3, and LCN2. This finding may partially explain the beneficial effects of exercise, caloric restriction, and ketogenic diets. Future studies will further investigate the mechanisms through which 2-DG is reducing astrocyte inflammatory gene expression as well as the downstream effects on neuronal functioning and blood–brain barrier integrity.

## Data Availability

All data will be available by reasonable request.

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
