# Peer review of "Caloric Restriction Mimetic 2-Deoxyglucose Reduces Inflammatory Signaling in Human Astrocytes: Implications for Therapeutic Strategies Targeting Neurodegenerative Diseases"

_brainsci, 2022, doi:10.3390/brainsci12030308_

Round 1
Reviewer 1 Report
The present data suggest that inhibiting glycolysis in human astrocytes reduces IL-1β-induced reactivity. This finding may lead to novel therapeutic strategies to limit inflammation and enhance bioenergetics toward the goal of preventing and treating neurodegenerative diseases. However, the present work does not address the possible toxicity of 2DG on astrocytes.
Major Questions: In addition to blocking glycolysis, 2DG has also been shown to inhibit N-linked glycosylation in the endoplasmic reticulum (ER) by interfering with the addition of mannose to the oligosaccharides. 2DG inhibition of N-linked glycosylation is a significant contributor to the toxicity in multiple cell lines. Interference with N-linked glycosylation may result in improper protein folding, leading to ER stress and the activation of the unfolded protein response (UPR), and if the stress is sufficiently severe, it may lead to the initiation of cell death pathways. Therefore, it is necessary to show the cell viability of astrocyte cultures upon 2DG at the concentrations studied (1-100 mM) in 24 h of incubation.
Also, it is necessary to describe in Methods section the sequences (forward- and reverse sense) of the various primers used to quantify the expression of the genes studied (IL-6, IL-IB, TNF, C3 and LCN2) by RT2-PCR.
Author Response
R1 comment 1: “The present work does not address the possible toxicity of 2DG on astrocytes”
Author Response: Thank you for the astute observation that 2DG can be cytotoxic and this should be ruled out as a cause for the observed reduction in inflammatory gene expression in human astrocytes. To test cytotoxicity of 2DG, we treated astrocyte cultures with increasing doses of 2DG as in figure 1 of the original manuscript and then stained and counted nuclei and ran tested for toxicity using the CytoTox Glo assay from Promega and illustrated in Figure 4. As now illustrated in Figure 4, 2DG had no cytotoxic effect on human astrocytes. The new methods and data are described in blue text in lines 22, 195-211, 275-284, 287-293, and 363-368.
R1 comment 2: It is necessary to describe in Methods section the sequences (forward- and reverse sense) of the various primers used to quantify the expression of the genes studied (IL-6, IL-IB, TNF, C3 and LCN2) by RT2-PCR.
Author Response: The authors agree that it is necessary to have access to the primers for the real-time pcr assays. Fortunately, all of the assays we used are commercially available from ThermoFisher and the RefSeq is available on their website. We have included in the methods the catalog numbers for each taqman assay used. (Lines 155-157)
Reviewer 2 Report
Plaease revised the text. Some corrections to minor methodological errors and text editing are required. Ony one question about the method: Has the use of tissues from fetal human brain tissue from elective terminated pregnancy been approved by an ethics committee?
Author Response
R2 comment 1: “ Some corrections to minor methodological errors and text editing are required.”
Author Response: Thank you. We have revised the text to correct any errors.
R2 comment 2: “Has the use of tissues from fetal human brain tissue from elective terminated pregnancy been approved by an ethics committee?”
Author Response: We have included the UCSD Human Research Protections Program and IRB numbers and dates in the Methods in Lines 106-109.
Round 2
Reviewer 1 Report
The revised version of manuscript has been sufficiently improved to warrant publication in Brain Sciences.